# Word-Level Emotional Expression Control in Zero-Shot Text-to-Speech Synthesis

**Tianrui Wang**[1,2,3], **Haoyu Wang**[1], **Meng Ge**[1], **Cheng Gong**[4], **Chunyu Qiang**[1,5],
**Ziyang Ma**[3,6], **Zikang Huang**[1], **Guanrou Yang**[6], **Xiaobao Wang**[1,2],
**Eng Siong Chng**[3], **Xie Chen**[6], **Longbiao Wang**[1,7]*, **Jianwu Dang**[8]

[1]Tianjin Key Laboratory of Cognitive Computing and Application, College of
Intelligence and Computing, Tianjin University, [2]Guangdong Laboratory of Artificial
Intelligence and Digital Economy (SZ), [3]Nanyang Technological University,
[4]TeleAI, China Telecom, [5]Kuaishou Technology, [6]Shanghai Jiao Tong University,
[7] Huiyan Technology (Tianjin), [8]Shenzhen Institute of Advanced Technology

## Abstract

While emotional text-to-speech (TTS) has made significant progress, most existing research remains limited to utterance-level emotional expression and fails to support word-level control. Achieving word-level expressive control poses fundamental challenges, primarily due to the complexity of modeling multi-emotion transitions and the scarcity of annotated datasets that capture intra-sentence emotional and prosodic variation. In this paper, we propose WeSCon, the first self-training framework that enables word-level control of both emotion and speaking rate in a pretrained zero-shot TTS model, without relying on datasets containing intra-sentence emotion or speed transitions. Our method introduces a transition-smoothing strategy and a dynamic speed control mechanism to guide the pretrained TTS model in performing word-level expressive synthesis through a multi-round inference process. To further simplify the inference, we incorporate a dynamic emotional attention bias mechanism and fine-tune the model via self-training, thereby activating its ability for word-level expressive control in an end-to-end manner. Experimental results show that WeSCon effectively overcomes data scarcity, achieving state-of-the-art performance in word-level emotional expression control while preserving the strong zero-shot synthesis capabilities of the original TTS model.

## 1 Introduction

Humans possess the ability to regulate emotional expression during speech flexibly [1]. To simulate this expressive capability, recent advances in text-to-speech synthesis (TTS) have increasingly focused on controllable generation of various aspects of speech, such as timbre, emotion, and speaking rate [2]. Such control is a key objective in the development of human-like and expressive TTS.

Most current TTS models exhibit zero-shot capabilities, enabling them to synthesize speech from text while cloning attributes such as timbre, emotion, and speaking rate from a reference speech sample [3, 4, 5]. Despite these advances, as shown in Figure 1, emotional and speaking rate control in current models is typically limited to the utterance level. This differs significantly from how humans naturally express emotion in speech. **Unlike global speaker identity, emotional expression and speaking rate are dynamic and often vary within a single sentence** [6, 7]. Therefore, word-level control of these factors is essential for achieving more natural and expressive speech synthesis [8]. To address this limitation, some approaches have proposed phoneme-level emotion prediction from target

---

*Corresponding Author.

39th Conference on Neural Information Processing Systems (NeurIPS 2025).

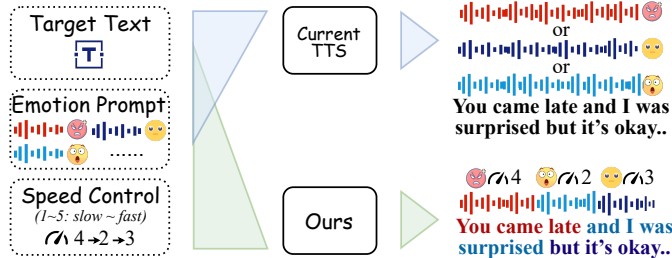

Figure 1: Word-level control of emotion and speaking rate aims to modulate both attributes within an utterance, guided by multiple emotional prompts and emotion-speed-tagged text. Our approach, WeSCon, achieves this using only a small-scale public dataset without emotion transitions.

text to guide expressive synthesis [9, 10, 11]. While these methods show potential for word-level emotion control, relying solely on text makes it difficult to capture essential acoustic cues such as prosody and intensity, which are vital to emotional expression control [12, 13, 14]. To address this limitation, recent studies such as ELaTE [15] and EmoCtrl-TTS [16] have demonstrated that reference speech with emotional content can support intra-utterance control of time-varying expressive patterns, such as transitions from laughter to crying. These works reflect a growing interest in TTS with word-level control over both emotion and speaking rate, but they also underscore several fundamental challenges. First, word-level expression control requires multiple emotional speech prompts, which introduces the challenge of guiding the model to attend to the appropriate emotion at each word. In addition, current methods for fine-grained expression control rely on large-scale emotional speech datasets with time-aligned emotion transitions. However, such datasets are limited in both scale and accessibility [17], making fine-grained control even more difficult to realize in practice. These challenges lead us to ask: **Is it possible to achieve effective word-level control of both emotion and speaking rate without relying on speech datasets containing emotion or speed transitions?**

In this work, motivated by the zero-shot potential of pretrained TTS models, we propose WeSCon, a two-stage self-training framework that achieves **W**ord-level **E**motion and **S**peed **Con**trol for TTS using only a small amount of public speech data without emotion or speed transitions. In the first stage, we design a multi-round inference framework that incorporates a transition-smoothing module and a dynamic speed control mechanism. Without relying on any emotional training data, this approach enables a pretrained zero-shot TTS model to perform high-quality word-level emotional expression control in TTS. In the second stage, the original TTS model is repurposed as a student and trained under the supervision of the 1st-stage teacher. A dynamic emotional attention bias is introduced, enabling the student to acquire word-level control of emotion and speed through a simplified end-to-end inference process, without the need for complex iterative generation or smoothing. Experimental results show that WeSCon achieves state-of-the-art performance on the task of word-level emotional expression control in TTS, while preserving the zero-shot generalization and generation capabilities of the pretrained TTS model. Our contributions are summarized as follows:

- We propose a multi-round inference mechanism equipped with transition smoothing and dynamic speaking rate control, which is the first to achieve word-level control of both emotion and speaking rate in TTS without relying on any emotional training data.

- We further introduce a novel self-training framework with a dynamic emotional attention bias mechanism that empowers a pretrained TTS model with end-to-end word-level emotion and speaking rate control, using limited data without intra-sentence emotion or speed transitions.

- We conduct comprehensive experiments to validate the effectiveness of our proposed framework. Results show that our method enables a pretrained zero-shot TTS model to achieve SOTA performance in word-level emotional expression control, while preserving its original zero-shot capabilities. Ablation studies further confirm the contribution of each key design component. Our samples are available at `https://wangtianrui.github.io/wescon/`.

## 2   Related Work

**Scarcity of Emotional Dataset**   The development of controllable TTS, particularly for emotional expression control, depends heavily on high-quality emotional speech datasets [18, 19, 20]. While

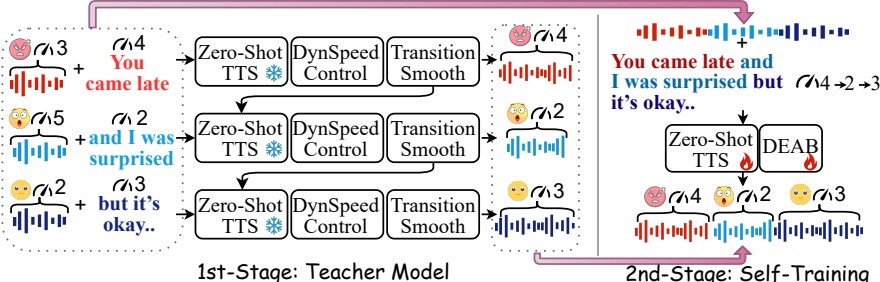

Figure 2: Overview of WeSCon. The 1st-stage teacher extends a zero-shot TTS model with dynamic speed control, transition smoothing, and multi-round inference to enable word-level emotion and speaking rate control. In the 2nd stage, it supervises a student model with a dynamic emotion attention bias (DEAB) to achieve the same control in an end-to-end manner with reduced inference complexity.

public corpora such as ESD [21], IEMOCAP [22], and CREMA-D [23] are available, they primarily provide utterance-level annotations and lack word-level or time-aligned emotional labels. These datasets are also limited in size and diversity, often consisting of scripted speech and covering a narrow range of emotions and speakers [24]. More importantly, emotional datasets with intra-sentence variation, which are essential for learning word-level control, remain extremely scarce and are typically restricted to private use [15]. Creating such datasets is expensive, requiring detailed word- or frame-level annotation and subjective emotional labeling [25]. This lack of fine-grained emotional data poses a major challenge for training models capable of word-level expressive TTS.

**From Utterance-Level to Word-Level Controllability of Emotion and Speaking Rate** Most controllable TTS systems support only utterance-level control, where a single label or reference speech governs the entire sentence [26, 27]. To achieve word-level control, some methods attempt to predict frame- or phoneme-level emotional indicators from text alone [28, 29, 30], but they often fail to capture expressive variability due to the lack of acoustic cues such as intensity and prosody [9, 10, 11]. Other approaches, such as ELaTE [15] and EmoCtrl-TTS [16], introduce emotional reference speech to enable intra-utterance control of specific expressive patterns like laughter or crying. While these represent progress, they are typically limited in expressiveness or rely on large-scale emotional datasets that are rarely publicly available. Consequently, achieving general and flexible word-level control over both emotion and speaking rate remains a major challenge.

**Self-Training under Data Scarcity** Self-training has become a promising approach for low-resource speech signal processing, enabling knowledge transfer without fine-grained datasets [31, 32]. While it has been applied to tasks like speaker adaptation [33], paralinguistic modeling [34], and speech translation [35, 36], its use for fine-grained emotional control in TTS remains unexplored, especially without detailed expressive labels. To address the scarcity of fine-grained datasets for word-level expressive control, we propose a self-training framework where a teacher model with multi-round inference, transition smoothing, and dynamic speed control generates expressive pseudo-labels. A student model, sharing the teacher's backbone, is then fine-tuned under its supervision to perform word-level emotion and speaking rate control through a simplified end-to-end inference process, using only a small public dataset without intra-sentence emotion or speed transitions.

## 3 WeSCon

### 3.1 Overview

WeSCon is a two-stage self-training framework that enables word-level control of emotion and speaking rate in a pretrained zero-shot TTS model, using only a small amount of emotional speech data without intra-sentence emotion transitions as prompts. As shown in Figure 2, in the first stage, we introduce a multi-round inference process with transition smoothing and dynamic speaking rate control to generate speech with word-level expression variations. In the second stage, the 1st-stage model acts as a teacher to guide the original TTS model, equipped with a dynamic emotional attention bias (DEAB), toward word-level control through a simplified end-to-end inference. Sections 3.2 and 3.3 describe the two stages, and Section 3.4 provides the training details.

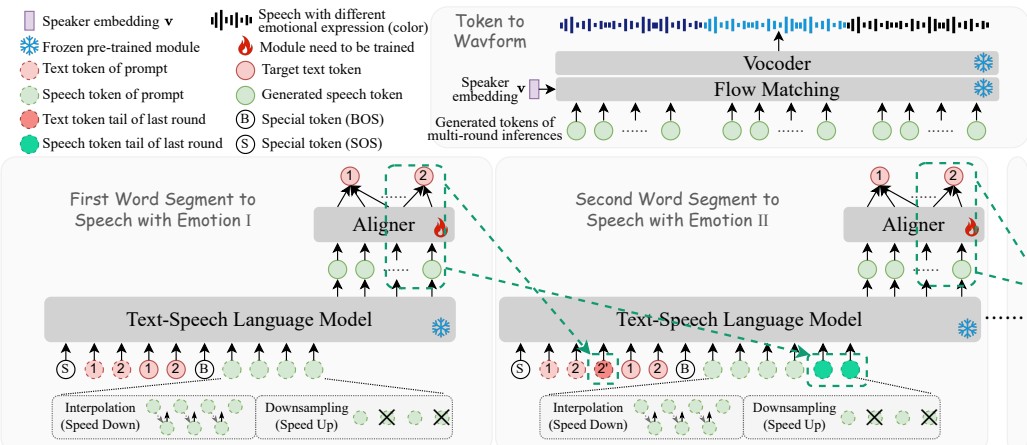

Figure 3: Word-level emotion and speaking rate control using a transition-smoothing module and dynamic speed adjustment. At each inference round, an emotional prompt is used to generate a speech segment, with the tail of the previous output appended to ensure continuity. Speaking rate is controlled by interpolating or downsampling prompt speech tokens. The final utterance is produced by concatenating all segments and decoding them through flow matching and a vocoder.

## 3.2 Teacher Model

### 3.2.1 Word-Level Emotion Control

As discussed in Section 2, current TTS models can perform utterance-level emotion and speaker cloning. Building on this, we adopt the high-performance CosyVoice2 [37] as our backbone (details of the backbone architecture are provided in Appendix A) and propose a multi-round inference strategy, where the model synthesizes multiple segments using different emotional prompts to achieve word-level emotion control. While this approach enables flexible emotional modulation, it often causes unnatural acoustic discontinuities at segment boundaries. To address this, we introduce a transition-smoothing mechanism that improves coherence across inference rounds, as illustrated in Figure 3. Without modifying CosyVoice2, we append a lightweight content aligner, composed of non-causal Transformer [38] and convolutional layers. Trained on ASR data, this module predicts the corresponding text token for each speech token and requires no emotional supervision. During inference, the input text is segmented based on a user-defined emotion plan. At each inference round, the final text and speech tokens from the previous round are appended to the current prompt, forming an explicit tail-to-head linkage. This aligns naturally with CosyVoice2's continuation-style generation [39, 40], enabling smooth and coherent emotional transitions.

### 3.2.2 Word-Level Speaking Rate Control

In CosyVoice2, utterance-level temporal prosody, including speaking rate and duration, is entirely determined by the reference speech prompt. To support more flexible and word-level control of speaking rate within a single utterance, we introduce a dynamic speed control mechanism as part of our multi-round inference framework, as illustrated in Figure 3. The core idea is to adjust the prompt speech tokens using either nearest-neighbor interpolation or downsampling. Interpolation extends the prompt length, which slows down the generated speech, while downsampling shortens the prompt, resulting in a faster speaking rate. As demonstrated in Appendix B, this resampling method provides effective global prosody control. By integrating it into the multi-round inference process, the speaking rate can be dynamically controlled at the word level as needed.

### 3.2.3 Speaker Consistency

Although the speech tokens in CosyVoice2's language model (LM) are primarily designed to encode semantic information (as introduced in Appendix A), these speech tokens may still inadvertently leak a small amount of speaker-related information. In contrast, the flow matching serves as a voice conversion-based reconstructor that transforms the generated speech tokens into the voice of a

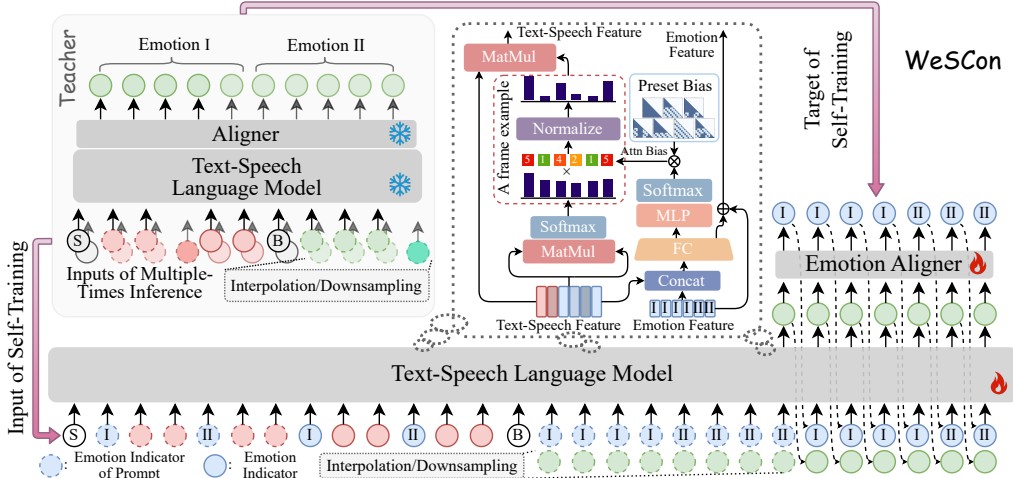

Figure 4: The proposed self-training strategy. A teacher model under a complex multi-round inference manner supervises a student TTS model to enable word-level emotion and speaking rate control. The dynamic emotional attention bias mechanism further enhances expressive generation in a simplified end-to-end single-pass inference manner.

specified target speaker. This design implies that as long as speaker inconsistency is avoided during the multi-round inference process in the LM part, the flow matching can effectively enforce speaker consistency in the final output. To ensure this consistency, we adopt a speaker-aware prompt selection strategy. Specifically, during multi-round inference, we prioritize selecting emotional prompts from different emotions of the same speaker. Then, a reference sample from the target speaker is randomly selected to provide the speaker identity to flow matching for generating the target speaker's speech.

## 3.3 Self-Training

In the previous section, we enabled word-level control of emotion and speaking rate by introducing a multi-round inference framework for CosyVoice2 [37]. However, components such as the non-causal content aligner, multi-round inference, and tail-to-head linkage introduce significant inference complexity. To reduce this overhead while preserving controllability, we adopt a self-training strategy. As shown in Figure 4, the enhanced first-stage model serves as a teacher to supervise the original TTS model. The student model, equipped with a dynamic emotional attention bias, learns to achieve word-level emotion and speaking rate control through a simplified end-to-end inference.

### 3.3.1 Self-Training with Teacher-Generated Emotion-Transition Speech

Our teacher model achieves word-level control of emotion and speaking rate without modifying the original TTS parameters, relying instead on a complex inference pipeline with dynamic speed control and multi-round generation. To transfer this fine-grained control ability to a simplified end-to-end model, we propose a self-training strategy. Specifically, the 1st-stage teacher model guides the student model to learn word-level controllability. We first use GPT-4o [41] to generate emotion-transition text sequences (details are shown in Appendix D), which are paired with public emotional speech samples (without emotion transitions) as prompts. The teacher then synthesizes speech with word-level variation in emotion and speaking rate. These outputs are filtered based on character accuracy and expressive similarity (details are introduced in Appendix E), and the student model is fine-tuned on the filtered supervisions with a small learning rate. This enables word-level emotional expression control during inference without requiring multi-round generation or dynamic concatenation.

### 3.3.2 Dynamic Emotional Attention Bias

We aim to preserve the strong zero-shot capability of the original TTS model while enabling word-level control of emotional expression under the self-training framework. To achieve this, we formulate the input structure as $\{Ⓢ, C^{\text{prompt I}}, C^{\text{prompt II}}, \ldots, C^{\text{tgt}}, Ⓑ, S^{\text{prompt I}}, S^{\text{prompt II}}, \ldots, S^{\text{tgt}}\}$, where

$\boldsymbol{C}^{\text{prompt } i}$ and $\boldsymbol{S}^{\text{prompt } i}$ denote the text and speech tokens of the $i$-th emotional prompt, respectively. $\boldsymbol{C}^{\text{tgt}}$ is the target text token sequence, and $\boldsymbol{S}^{\text{tgt}}$ is the corresponding speech token sequence used as supervision. The symbols ⑤ and ⑧ indicate the beginning of text and speech. This design remains fully compatible with the original input format $\{⑤, \boldsymbol{C}, ⑧, \boldsymbol{S}\}$ of CosyVoice2, preserving the autoregressive pattern of the pretrained model. To further encode word-level emotional variation within this unified format, we extend the text-side input by inserting explicit emotion indicator tokens that mark the boundaries between emotional segments. As illustrated in Figure 4, the final input sequence preceding ⑧ becomes $\{⑤, E^{\text{I}}, \boldsymbol{C}^{\text{prompt I}}, E^{\text{II}}, \boldsymbol{C}^{\text{prompt II}}, \dots\}$, where each $E^i$ acts as a soft anchor guiding the model to modulate emotion transitions during generation.

While the above data formatting preserves CosyVoice2's generalization by avoiding interference with learned knowledge, it introduces a new challenge: during synthesis, the model may incorrectly attend to emotion-inconsistent prompts. For instance, when generating speech aligned with Emotion I, attention may drift toward prompts labeled with Emotion II, leading to emotional inconsistency and degraded synthesis quality. To address this, we propose a dynamic attention bias mechanism that constrains the model's focus to emotion-relevant prompt regions based on the predicted emotional trajectory. Concretely, we introduce a causal lightweight Transformer to predict token-level emotion labels $E_t^{\text{tgt}}$ for each speech token $\boldsymbol{S}_t^{\text{tgt}}$ from historical context. Using the predicted emotion sequence, we introduce a dynamic attention bias mechanism at each Transformer layer. We first concatenate the current text-speech representation with the predicted emotion features and project it through a linear layer. The output is processed in two ways: one path adds a residual and feeds into the next layer, while the other is passed to an MLP [42] and softmax to produce a weight vector $\boldsymbol{\omega} \in \mathbb{R}^{1 \times 7}$. The $\boldsymbol{\omega}$ is then used to compute a dynamic attention bias by linearly combining seven predefined attention bias templates $\boldsymbol{B}^{\text{temp}} \in \mathbb{R}^{7 \times T \times T}$ (see Appendix F for details). The resulting bias is computed as:

$$\boldsymbol{B}^{\text{bias}} = \sum_{i=0}^{6} \omega_i \cdot \boldsymbol{B}_i^{\text{temp}}. \tag{1}$$

Then we multiply the bias with the softmax-normalized attention to selectively emphasize regions aligned with the current emotional context. The final self-attention output is computed as:

$$\boldsymbol{O} = \left( \frac{\text{Softmax}\left(\frac{\boldsymbol{Q}\boldsymbol{K}^{\top}}{\sqrt{d}}\right) \odot \boldsymbol{B}^{\text{bias}}}{\sum_{j=1}^{T} \left[\text{Softmax}\left(\frac{\boldsymbol{Q}\boldsymbol{K}^{\top}}{\sqrt{d}}\right) \odot \boldsymbol{B}^{\text{bias}}\right]_{:,j}} \right) \boldsymbol{V}, \tag{2}$$

where $\boldsymbol{Q}, \boldsymbol{K}, \boldsymbol{V} \in \mathbb{R}^{H \times T \times d}$ denote the multi-head ($H$) query, key, and value, respectively, and $d$ is the attention head dimension. The operator $\odot$ denotes element-wise multiplication. This formulation enables the model to dynamically focus on emotionally relevant prompt segments at each generation step, thereby improving alignment between the generation and the intended emotional trajectory.

### 3.4 Detail Training Setup

WeSCon is trained in two stages. The first stage trains a content aligner to ensure smooth transitions during multi-round inference. In the second stage, a self-training strategy is adopted to transfer the teacher model's ability to control word-level emotional expression to the original TTS model.

**The First Stage (Teacher Model)**  We use forced alignment [43] to generate token-level alignments between transcripts and speech, which serve as supervision for the content aligner. The TTS model remains frozen throughout this stage. Training of the content aligner is conducted without multi-round forwards. Let $\boldsymbol{C}$ and $\boldsymbol{S}$ denote the input text and speech token sequences, $\boldsymbol{Y}^{\text{token}} \in \mathbb{N}^T$ denote the aligned target token sequence, where each label corresponds to one of $V_1$ token classes. Let $\boldsymbol{Y}^{\text{bd}} \in \mathbb{R}^{T \times 1}$ be the binary label sequence for content boundary detection. The content aligner is jointly trained with a token-level content classification loss and a binary boundary detection loss:

$$\mathcal{L}_{\text{aligner}} = -\sum_{t=T^{\text{C}}}^{T^{\text{S}}-1} \log p\left(Y_t^{\text{token}} \mid ⑤, \boldsymbol{C}, ⑧, \boldsymbol{S}; \theta^{\text{tts}}, \theta^{\text{ca}}\right) - \log p\left(Y_t^{\text{bd}} \mid ⑤, \boldsymbol{C}, ⑧, \boldsymbol{S}; \theta^{\text{tts}}, \theta^{\text{ca}}\right), \tag{3}$$

where $T^{\text{C}}$ and $T^{\text{S}}$ denote the last frame indices for text and speech, and $T = T^{\text{S}} - T^{\text{C}}$ is the total number of speech tokens. The learnable parameters $\theta^{\text{ca}}$ correspond to the content aligner, while $\theta^{\text{tts}}$ is the frozen TTS model parameter used during forward propagation. We also apply class weighting

during loss computation to reduce the impact of overrepresented silence tokens and address the imbalance in boundary label distribution [44].

**The Second Stage (Self-Training)** The teacher model generates supervision via multi-round inference using GPT-4o-generated texts with emotion labels. Token-level emotion labels are aligned based on emotion-text correspondence. The student model is optimized by two objectives. The first is a negative log-likelihood for speech token prediction:

$$\mathcal{L}_{\text{tts}} = -\sum_{t=T^{\text{prompt}}}^{T^{\text{tgt}}-1} \log p\left(S_t^{\text{tgt}} \mid \text{\textcircled{S}}, \boldsymbol{C}^{\text{prompt}}, \boldsymbol{C}^{\text{tgt}}, \boldsymbol{E}^{\text{text}}, \text{\textcircled{B}}, \boldsymbol{E}_{<t}^{\text{speech}}, \boldsymbol{S}^{\text{prompt}}, \boldsymbol{S}_{<t}^{\text{tgt}}; \theta^{\text{tts}}, \theta^{\text{ea}}\right), \quad (4)$$

where $\boldsymbol{C}$ and $\boldsymbol{S}$ are text and speech tokens for prompt and target, $\boldsymbol{E}$ are text-level and token-level emotion labels, and trainable $\theta^{\text{tts}}, \theta^{\text{ea}}$ denote TTS model and emotion aligner parameters. The second is a token-level cross-entropy loss for emotion prediction:

$$\mathcal{L}_{\text{e}} = -\sum_{t=T^{\text{prompt}}}^{T^{\text{tgt}}-1} \log p\left(E_t^{\text{tgt}} \mid \text{\textcircled{S}}, \boldsymbol{C}^{\text{prompt}}, \boldsymbol{C}^{\text{tgt}}, \boldsymbol{E}^{\text{text}}, \text{\textcircled{B}}, \boldsymbol{E}_{<t}^{\text{speech}}, \boldsymbol{S}^{\text{prompt}}, \boldsymbol{S}_{<t}^{\text{tgt}}; \theta^{\text{tts}}, \theta^{\text{ea}}\right). \quad (5)$$

# 4 Experiments

## 4.1 Experimental Setup

**Data and Model Configuration** In the first stage, the content aligner is trained on 200 hours of non-emotional English-Chinese speech from LibriSpeech-100-Clean [45] and AISHELL-1 [46]. In the second stage, the teacher model uses non-transition emotional train-set from ESD [21] as prompts to synthesize training samples based on emotion-transition texts generated by GPT-4o (see Appendix D for generation details and examples). We adopt CosyVoice2 [37] as the backbone TTS model. The content aligner is composed of five non-causal Transformer layers and two 5×5 convolutional layers with stride 1 and batch normalization [47], following CosyVoice2's configuration for architectural consistency. In the second stage, the emotion aligner is a lightweight two-layer causal Transformer. The emotional attention bias module includes a linear layer with a hidden dimension of 14 and an MLP output dimension of 7.

**Setup of Training and Inference** In the first stage, the content aligner is trained for 400k steps on 2 NVIDIA 3090 GPUs using Adam [48] with a learning rate linearly warmed up to 2.5e-4 over the first 10% of steps, then linearly decayed to 0. Each batch contains 90 seconds of speech. In the second stage, the student model is trained for 600k steps on 4 NVIDIA 3090 GPUs. The TTS model is frozen for the first 20k steps to focus on training the emotion aligner. Each batch contains 40 seconds of speech, and Adam is used with a fixed learning rate of 5e-7. Repetition-aware top-$k$ sampling [49] is applied during inference, with $k = 50$ and temperature $= 0.9$.

**Evaluation** To evaluate word-level control over emotion and speaking rate, we construct test sets based on test set of ESD and use outstanding zero-shot TTS models [50, 51, 52, 37] with multi-round concatenative inference as baselines (see Appendix G for details). We use objective and subjective metrics to assess system performance (see Appendix G.3 for details). For intelligibility, we report WER using Whisper-Large [53] for English and CER using Paraformer [54] for Chinese. Speaker similarity (S-SIM) is computed via cosine similarity of WavLM-Large embeddings [55]. To evaluate prosody alignment, we use AutoPCP [56]. Emotion similarity metrics (Emo2v. and Aro.) are computed using emotion2vec-Large [57] and a wav2vec-based model [58], respectively. We use the variance of DNSMOS-Pro [59] (DNSV) to assess the naturalness of emotion transition. Subjective evaluation includes four kinds of Mean Opinion Score (MOS): SMOS (speaker similarity), NMOS (naturalness of emotion transition), EMOS (emotion match), and SPMOS (speed match), each rated on a 5-point scale. Both the mean and 95% confidence intervals of MOS are reported.

## 4.2 Experimental Results

### 4.2.1 Comparison with Reference Models

**Objective Evaluation** We evaluate our method on word-level emotion and speaking rate control in both English and Chinese TTS. As shown in Table 1, WeSCon (1st-stage) and WeSCon (2nd-stage) consistently outperform baselines on expressive metrics. Notably, the 2nd-stage model achieves

Table 1: Objective results on English and Chinese test sets for TTS with word-level emotion and speaking rate control. The best results for each metric are in **bold**, and the second-best are underlined.

| | Method | WER/CER↓ | DNSV↓ | S-SIM↑ | AutoPCP↑ | Emotion↑ Emo2v. | Aro. |
|---|---|---|---|---|---|---|---|
| English | Index-TTS | **2.611** | 8.967 | 0.387 | 2.436 | 0.858 | 0.434 |
| | F5-TTS | 2.954 | 8.972 | 0.453 | 2.417 | 0.869 | 0.447 |
| | Spark-TTS | 2.787 | 8.637 | 0.374 | 2.560 | 0.861 | 0.440 |
| | CosyVoice2 | 3.185 | 7.894 | 0.521 | 2.525 | 0.866 | 0.446 |
| | WeSCon (1st) | 3.204 | 4.577 | 0.531 | 2.689 | 0.879 | 0.463 |
| | WeSCon (2nd) | 3.192 | **4.361** | **0.532** | **2.707** | **0.882** | **0.468** |
| Chinese | Index-TTS | **1.834** | 8.521 | 0.490 | 2.470 | 0.838 | 0.514 |
| | F5-TTS | 1.965 | 9.134 | 0.478 | 2.541 | 0.847 | 0.510 |
| | Spark-TTS | 1.897 | 8.633 | 0.441 | 2.518 | 0.848 | 0.530 |
| | CosyVoice2 | 2.119 | 7.612 | 0.581 | 2.514 | 0.843 | 0.537 |
| | WeSCon (1st) | 2.129 | 4.980 | 0.595 | 2.650 | 0.866 | 0.551 |
| | WeSCon (2nd) | 2.122 | **4.210** | **0.599** | **2.663** | **0.872** | **0.556** |

the highest Emo2V. and Aro. scores in both languages, demonstrating strong word-level emotional expressiveness enabled by our self-training framework. Regarding transition smoothness, our models significantly reduce DNSV compared to CosyVoice2, with values dropping from 7.894 to 4.361 in English and from 7.612 to 4.210 in Chinese. This highlights the effectiveness of our smoothing mechanism and the end-to-end continuous inference in the 2nd-stage model in mitigating acoustic discontinuities across transitions. While the character error rate is slightly higher than baselines, it remains comparable to CosyVoice2, our backbone model. Finally, the 2nd-stage model slightly surpasses the 1st-stage model, benefiting from self-training with selective filtering that retains high-quality supervision from the teacher. Overall, our approach consistently improves upon CosyVoice2 and achieves SOTA performance in key aspects of word-level expressive controllable TTS.

**Subjective Evaluation** We conduct subjective evaluations covering emotional expressiveness (EMOS), speaking rate control (SPMOS), speaker similarity (SMOS), and naturalness of emotion transition (NMOS), with details provided in Appendix G.3. As shown in Table 2, our method, WeSCon, consistently outperforms all baselines. It achieves more expressive and control-

Table 2: Subjective results evaluated by 15 listeners, with 95% confidence intervals computed from the t-test.

| Method | EMOS ↑ | SPMOS ↑ | SMOS ↑ | NMOS ↑ |
|---|---|---|---|---|
| Index-TTS | 3.51±0.19 | 3.50±0.21 | 3.06±0.23 | 2.97±0.25 |
| F5-TTS | 3.63±0.15 | 3.51±0.21 | 3.11±0.25 | 2.84±0.26 |
| Spark-TTS | 3.55±0.19 | 3.63±0.18 | 2.96±0.24 | 2.99±0.26 |
| CosyVoice2 | 3.61±0.17 | 3.56±0.20 | 3.54±0.25 | 3.29±0.23 |
| WeSCon | **3.70±0.17** | **3.89±0.18** | **3.96±0.19** | **3.93±0.20** |

lable speech while maintaining speaker identity, demonstrating effective word-level control in emotional expression. Additionally, WeSCon delivers more natural-sounding speech with smoother and more accurate speaking rate modulation.

**Capability on Zero-shot TTS** In addition to introducing word-level controllability, we evaluate the performance of our method on the standard zero-shot TTS task using the SEED test set (test-zh) [60]. As shown in Table 3, the WeSCon (1st) model yields results identical to CosyVoice2, as the backbone TTS is frozen during this stage. The 2nd-stage model also achieves comparable results. Together with the findings in Table 1, these

Table 3: Objective evaluation on standard zero-shot TTS performance using character error rate (CER) and speaker similarity (S-SIM).

| Method | CER ↓ | S-SIM ↑ |
|---|---|---|
| CosyVoice2 [37] | 1.45 | 0.748 |
| WeSCon (1st) | same with CosyVoice2 | |
| WeSCon (2nd) | 1.47 | 0.744 |

results demonstrate that our method enables word-level emotion and speaking rate control without significantly degrading the original zero-shot TTS performance of the pretrained model.

### 4.2.2 Ablation Study

**Transition-Smoothing Mechanism** We evaluate the impact of the transition-smoothing mechanism by removing the tail-to-head alignment during multi-round inference in the 1st-stage model. As shown in Table 4, removing this mechanism ("w/o smoothing") leads to a substantial increase in DNSV (from 4.980 to 7.568), indicating degraded smoothness between expressive transitions. Additionally, speaker

(S-SIM) and emotion similarity (Emo2V. and Aro.) drop notably, suggesting that the discontinuity negatively affects both emotional expression and speaker consistency. These results confirm that our smoothing strategy plays a crucial role in ensuring coherent segment transitions during generation.

**Speaking Rate Control** To examine the effectiveness of our dynamic speaking rate control, we remove this component from the 1st-stage model ("w/o speed control"). As shown in Table 4, DNSV slightly increases from 4.980 to 5.067, and performance drops are observed across most expressive metrics, such as AutoPCP (2.650 to 2.499) and Emo2v. (0.866 to 0.844). This suggests that speaking rate variation provides important prosodic cues for emotional expression in TTS. In addition, we further investigate the interaction between speaking rate control and emotional expression in Appendix C.

Table 4: Ablation study on two stages for word-level controllability on Chinese testset.

| Method | CER↓ | DNSV↓ | S-SIM↑ | Auto PCP↑ | Emotion↑ Emo2v. | Aro. |
|---|---|---|---|---|---|---|
| WeSCon (1st) | 2.129 | **4.980** | **0.595** | **2.650** | **0.866** | **0.551** |
| w/o smoothing | 2.209 | 7.568 | 0.576 | 2.596 | 0.851 | 0.531 |
| w/o speed control | **2.126** | 5.067 | 0.582 | 2.499 | 0.844 | 0.526 |
| WeSCon (2nd) | **2.122** | **4.210** | **0.599** | **2.663** | **0.872** | **0.556** |
| w/o attention bias | 2.398 | 5.534 | 0.575 | 2.511 | 0.837 | 0.519 |
| w/o emotion flag | 2.455 | 5.880 | 0.573 | 2.492 | 0.831 | 0.515 |
| w/o datafilter | 2.237 | 4.494 | 0.592 | 2.627 | 0.859 | 0.542 |
| w/o dataformat | 4.141 | 5.697 | 0.579 | 2.504 | 0.819 | 0.509 |

**Dynamic Emotional Attention Bias** In the 2nd-stage model, we evaluate the effect of removing the dynamic emotional attention bias ("w/o attention bias"). As shown in Table 4, this results in a clear performance drop across all metrics, especially emotion similarity. DNSV also increases, indicating reduced smoothness. The results confirm the importance of the attention bias module in enabling the 2nd-stage model to focus on the correct emotional prompt during inference.

**Data format of Self-training** We further investigate the importance of data formatting in self-training. As shown in Table 4, removing the emotion flags ("w/o emotion flag") results in performance drops across all metrics, indicating that these flags play a crucial role in signaling the locations of emotional shifts to the model. Furthermore, replacing our input data format with a naive one that simply concatenates prompts and targets ("w/o data format"), as $\{C^{\text{prompt I}}, ⑧, S^{\text{prompt II}}, \dots, C^{\text{tgt}}, ⑧, S^{\text{tgt}}\}$ leads to the most significant degradation in expressive metrics, including a sharp increase in CER from 2.166 to 4.141. These results suggest that aligning the data organization with the structure used during pretraining allows the model to better leverage its pre-trained knowledge.

**Self-Training Data Size** We evaluate the impact of training data size in the self-training process by varying the amount of synthetic speech used to fine-tune the 2nd-stage model. Metrics are normalized between 0 and 1. As shown in Figure 5, performance improves with more data and peaks at 500 hours. Beyond this point, metrics begin to decline. This trend is attributed to the limited variety of emotional categories and speaker identities in the ESD, which restricts expressive diversity and leads to overfitting when the data scale becomes overly redundant.

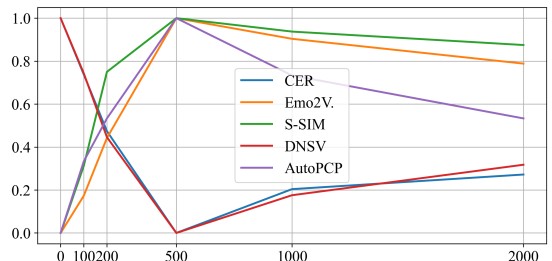

Figure 5: Performance trends on Chinese testset under different self-training data sizes.

**Out-of-Domain Generalization and Alignment** To further assess the model's robustness, we evaluate its generalization ability on out-of-domain data (Appendix H). In addition, we report the alignment accuracy achieved in both training stages (Appendix I).

## 5 Conclusion, Limitations, and Broader Impact

**Conclusion** In this paper, we propose WeSCon, the first method to overcome expressive data scarcity and enable word-level emotional expression control through end-to-end inference, under a self-training framework with a dynamic emotional attention bias mechanism. Experimental results show that WeSCon achieves state-of-the-art performance using only limited data without emotion or speed transitions, while maintaining strong zero-shot TTS capabilities.

**Limitations and Future Work**    1) Gradual emotion transitions. While WeSCon achieves smooth signal-level transitions, it lacks semantic modeling of emotional evolution. In human speech, emotional changes often involve intermediate states. 2) Emotion diversity and composition. The model is limited to a fixed set of discrete emotions and does not support compositional or blended expressions, such as combining anger and sadness to convey despair. 3) Conditioned control. Emotional transitions are currently predefined by GPT-4o-based plans, which restricts flexibility. Future work will explore more dynamic, context-aware control strategies to enable natural, interactive emotional expression.

**Broader Impact**    WeSCon can be applied to expressive speech synthesis, virtual agents, and emotional storytelling. However, it may also pose risks related to speaker impersonation, especially when specific content and speaker prompts are combined. Like other generative models, it may produce biased or inappropriate outputs, although no such cases were observed during testing.

## 6    Acknowledgements

This work was supported by the National Natural Science Foundation of China under Grant U23B2053 and Grant 62176182 and the Open Research Fund from Guangdong Laboratory of Artificial Intelligence and Digital Economy (SZ) (No.GML-KF-24-16).

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

# A  Details of CosyVoice2

CosyVoice2[2] [37] is a zero-shot TTS model based on a language model (LM) and flow matching. It first converts speech into discrete tokens through a supervised speech tokenizer module. Its core architecture is identical to the base structure illustrated in Figure 3, excluding the additional modules introduced in this work. The supervised speech tokenizer is jointly trained with an ASR task, which encourages the LM component to focus more on semantic modeling, particularly in terms of content, emotional expression, and duration. The flow matching component incorporates speaker embeddings[3] and target speech to provide speaker characteristics. It transforms the speech tokens produced by the language model into mel-spectrograms, primarily controlling global aspects of speech, especially speaker identity. Finally, the vocoder converts the mel-spectrograms into waveform signals. CosyVoice2's disentangled modeling of semantic content and speaker identity provides an important foundation for our method. In addition, since its training data is primarily in Chinese, it demonstrates significantly better performance in Chinese than in English.

# B  Speed Control

As described in Section 3, we control the speaking rate of synthesized speech by applying simple interpolation and downsampling to the prompt speech tokens. To assess whether this dynamic mechanism supports time-varying modulation, we visualize six types of control patterns in Figure 6. Numeric labels indicate the ratio between the transformed and original token lengths, where a ratio of 1 indicates no change, 0.5 indicates downsampling to half the length, and 2 represents interpolation that doubles it. The left panel illustrates three downsampling patterns: a gradually increasing interval, a decreasing interval, and a uniform interval. The right panel shows corresponding interpolation patterns. These results demonstrate that global interpolation/downsampling can produce effects comparable to time-varying interpolation/downsampling, particularly when accounting for the inherent randomness introduced by LM sampling. Because both methods provide only utterance-level control over speaking rate, word-level modulation requires integration with our multi-round inference framework.

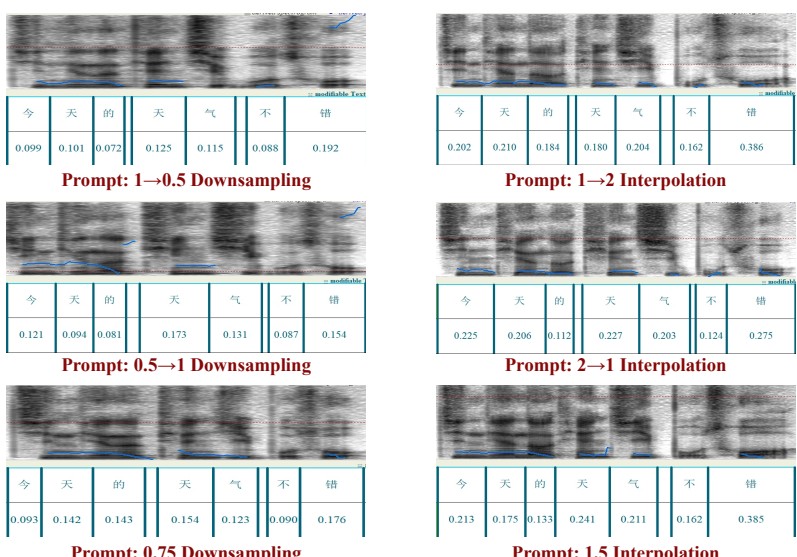

Figure 6: Visualization of six dynamic speaking rate control patterns, including time-varying and uniform interpolation/downsampling. The numerical labels indicate the ratio of the token length transformation relative to the original prompt. Blue numbers represent the duration (in frames) assigned to each character. All synthesized speech shares the same content, which is marked in blue text within the figure.

---

[2] https://huggingface.co/spaces/FunAudioLLM/CosyVoice2-0.5B
[3] https://github.com/alibaba-damo-academy/3D-Speaker/tree/main/egs/3dspeaker/sv-cam++

We further investigate how different resampling ratios influence the speaking rate of the generated speech. As shown in Figure 7, the results reveal a clear correlation between the resampling factor and the output speed. When the token length is reduced to less than 40% of the original through down-sampling, the model fails to produce intelligible speech, as indicated by the red circles. Conversely, interpolation beyond three times the original length has minimal additional effect on speaking rate. Notably, the most stable and effective control is achieved when the token length lies between 50% and 200% of the original, suggesting this range as a practical bound for reliable modulation.

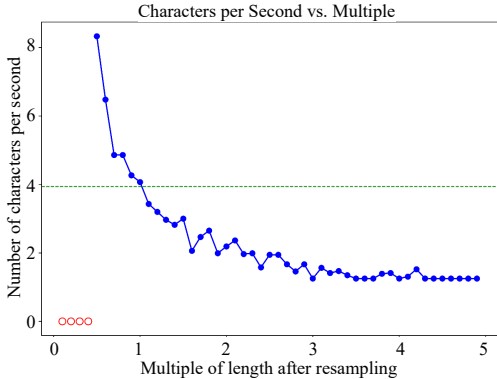

Figure 7: Correlation between resampling ratio and output speaking rate. The most effective control is observed when token lengths range from 50% to 200% of the original. Red circles mark failure cases where intelligibility is lost. The green dashed line indicates the character-per-second rate of the prompt audio.

## C   Interaction Between Speaking Rate and Emotion

This section further examines the relationship between speaking rate and emotional expression. Speaking rate and emotional state are strongly coupled in human speech, as different emotions are typically associated with distinct prosodic rhythms and energy patterns. Since both rate and emotional cues are derived from the same prompt speech, temporal resampling inevitably alters the perceived emotional expression. To quantify the effect of speaking rate on perceived emotion, 100 emotional utterances are randomly selected from the test set as prompts. The resampling ratio is systematically varied from 0.5 to 2.0 in increments of 0.25, using the same target text for all conditions, as summarized in Table 5. For each condition, the emotion similarity between the generated speech and both the original and rate-matched (re-rated) reference speech is calculated using the Emo2v. score.

Table 5: Effect of speaking rate variation on emotion similarity.

| Resampling Ratio | Emo2v. ↑ | Emo2v. (Re-rated) ↑ |
|---|---|---|
| 0.5 (downsampled to half, speed up) | 0.57 | 0.86 |
| 0.75 | 0.85 | 0.88 |
| 1.0 | 0.90 | 0.90 |
| 1.25 | 0.83 | 0.89 |
| 1.5 | 0.75 | 0.90 |
| 1.75 | 0.68 | 0.91 |
| 2.0 (interpolated to twice, speed down) | 0.51 | 0.87 |

The results show that emotion similarity declines substantially when the reference is not rate-matched, whereas it remains stable when compared with rate-adjusted references. Notably, when the resampling ratio deviates significantly from the natural range (e.g., 0.75 ~1.5), the perceived emotion becomes less consistent, likely due to distortion of spectral dynamics and pitch contours caused by excessive time-stretching or compression. These findings confirm that speaking rate provides essential prosodic cues for emotional perception, consistent with the observations in Section 4.2.2.

## D   Generation of Emotionally Varying Texts

We employ GPT-4o [41] to generate the corpus of sentences containing intra-sentence emotional transitions. To ensure the emotional transitions are contextually plausible, we construct prompts based on predefined scenarios, character relationships, and conversation topics. An example of the prompt is shown in Listing **??**. Specifically, we first create a large pool of randomly generated environments, contexts, and interpersonal relationships with personality traits. During generation, three elements are randomly selected and injected into the prompt to guide GPT-4o in producing scripts.

```
[caption={Example prompt for generating sentences with emotion shifts
    using GPT-4o.}, label={prompt}]{json}
You are a scriptwriter tasked with creating emotionally expressive **
    single-sentence dialogues with internal emotion shifts**. Your
    output should be grounded in the following:
- **Dialogue environment and external factors**,
- **Dialogue content and situational context**,
- **Interpersonal relationships and character traits**.
# Output Format Template
Each dialogue entry consists of **a list of sentence segments**, where
     **each segment is labeled with its corresponding emotion and
    speaking speed**. The entire list represents a single sentence
    spoken by a character.

Example:
[
    [
        {
            "lines_seg": "I trusted you",
            "emotion": "sad",
            "speed": "1.25"
        },
        {
            "lines_seg": "but you",
            "emotion": "surprise",
            "speed": "0.9"
        },
        {
            "lines_seg": "lied to me!",
            "emotion": "angry",
            "speed": "1.5"
        }
    ],
    ...
]
# Key Task Requirements
- There are {num_speakers} characters: {', '.join(speakers)}
- Dialogue alternates between speakers; **no speaker may speak twice
    in succession**
- Each sentence must be internally segmented (2~4 segments) and
    exhibit **clear emotion transitions**
- Each segment must include:
  - **lines_seg**: a span of 2~4 words, with punctuation only at the
      end of the last segment
  - **emotion**: the expressed emotion in this segment, chosen from: {
      emotions}
  - **speed**: the speaking rate for this segment (range: 0.5 to 2.0,
      where 0.5 = very fast, 1 = normal, 2.0 = very slow)
# Dialogue Environment and External Factors
{environment}
# Dialogue Content and Context
{context}
# Interpersonal Relationships and Character Traits
{character_traits}
```

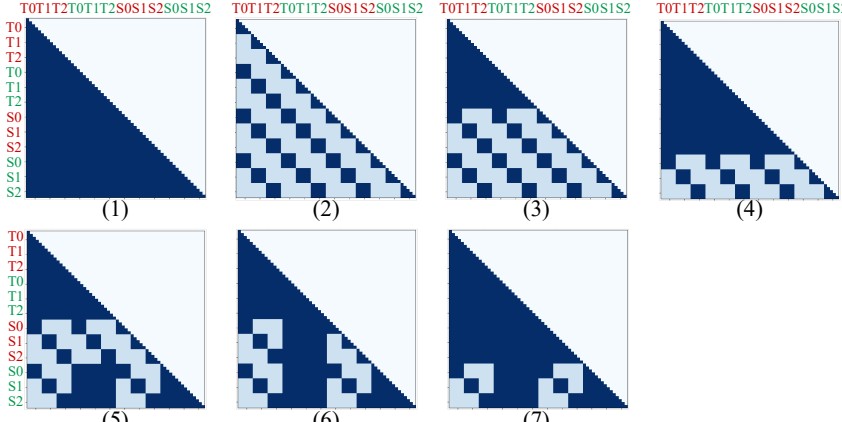

Figure 8: Illustration of seven predefined emotional attention bias patterns. Red elements denote prompt inputs, green elements denote target text and speech. **T** indicates text tokens, and **S** indicates speech tokens. Numbers represent emotional pairs indices. The light blue regions are preset to 1, the dark blue regions are preset to 5, and the upper-right triangular region is entirely set to 0.

## E Data Filtering in Self-Training

During the self-training process, we introduce a data filtering mechanism to ensure the reliability of the teacher model's guidance. Specifically, we adopt three metrics for evaluating the quality of generated speech: CER for Chinese and WER for English, speaker similarity, and emotion similarity. The first-stage teacher model has explicit access to the alignment among content, speech, emotion prompts, and speaker prompts, allowing us to directly compute these metrics with the prompt. To avoid introducing bias from the final objective evaluation metrics prematurely, we deliberately use models that differ from those employed during evaluation. For speech recognition, we adopt the SenseVoice model[4] [61]. For emotion representation, we use a Whisper model fine-tuned for speech emotion recognition[5]. For speaker embedding, we use Resemblyzer[6]. We normalize all three metrics for each data point and compute a combined score by summing them. Only the top 50% of data, ranked by this composite score, are selected for self-training. In other words, as shown in Figure 5, for a 500-hour training set, we actually generate approximately 1000 hours of data. Similarly, for a 2000-hour training set, we generate around 4000 hours of data.

## F Predefined Emotion Attention Bias

Since the emotional alignment of the student sequence input can be obtained from the output of the emotion aligner, we introduce seven predefined attention bias patterns to reduce the modeling burden of the emotional attention shift module. These typical patterns are illustrated in Figure 8, and described below.

(1) **Standard GPT-style Causal Attention.** Each token attends to all previous tokens in a standard autoregressive manner without any emotional constraints.

(2) **Strict Emotion-Aligned Attention.** This corresponds to the original training strategy of CosyVoice2. For instance, when decoding the second emotion segment (green S2), the model is only allowed to attend to the corresponding emotional prompt and its associated text, specifically red and green T2, and red S2.

(3) **Full Text History + Emotion-Aligned Speech Attention.** On top of (2), this setting allows text tokens to attend to the full text history, while speech tokens remain strictly aligned with their respective emotional segments.

---

[4]https://huggingface.co/FunAudioLLM/SenseVoiceSmall
[5]https://huggingface.co/firdhokk/speech-emotion-recognition-with-openai-whisper-large-v3
[6]https://github.com/resemble-ai/Resemblyzer

(4) **Full History Access for Prompt Speech Encoding.** Extending (3), this setting additionally allows each prompt speech token to access all previous tokens during encoding.

(5) **Prompt Speech Attends to Its Own History During Target Speech Generation.** During the generation of target speech, when prompt speech tokens are revisited, each token is allowed to attend to all previous prompt speech tokens.

(6) **Prompt Speech Self-Attention in Encoding.** Combining (4) and (5), this configuration allows prompt speech tokens to attend to the full history during encoding, but during target speech generation, they attend only to previously encoded prompt speech tokens.

Although some attention bias configurations, such as (5), (6), and (7), are relatively uncommon in standard architectures, our predefined template-based computation allows the Emotional Attention Bias module to focus solely on selecting and composing from these candidate biases. This design significantly reduces computational overhead and prevents the generation of implausible or inconsistent attention patterns.

## G  Evaluation Setup

### G.1  Details of Dataset

We use the train-set, dev-set, and test-set of ESD[7] [21] for training and evaluation. This dataset contains 350 parallel utterances, averaging 2.9 seconds in duration, spoken by 20 speakers: 10 native English and 10 native Mandarin (5 male and 5 female for each language). Each speaker expresses five emotions: happy, sad, neutral, angry, and surprised. All audio is sampled at 16 kHz.

For evaluation, we generate 1,000 emotion-speed-varying text samples (500 in Chinese and 500 in English) using the script provided in Appendix D. For each text sample, we randomly select emotional prompts from the ESD test set to match the emotion transitions required by the sentence. All emotion prompts within a single sentence are drawn from the same speaker to ensure consistency. The reference audio for the target speaker is also randomly selected from the same language-speaker subset. As a result, approximately 1 out of every 5 samples features emotional prompts and a target speaker from the same speaker-emotion setting, given that the ESD dataset contains 5 emotions.

### G.2  Baselines

We adopt four strong zero-shot TTS systems as baselines:

- **Index-TTS**[8] [50] is a GPT-style TTS model enhanced with pinyin-based pronunciation correction for Chinese characters and punctuation-based pause control. It integrates improved speaker condition modeling and BigVGAN2 [62] for high-quality audio synthesis. Trained on tens of thousands of hours of data, it supports multilingual zero-shot generation.

- **Spark-TTS**[9] [52] is a large language model-based TTS system built upon Qwen2.5 [63]. It directly reconstructs waveforms from LLM-predicted codes, eliminating the need for separate acoustic models. This design simplifies the pipeline and improves inference efficiency. It supports zero-shot voice cloning, cross-lingual/code-switching synthesis, and virtual speaker customization via controllable parameters such as gender, pitch, and speaking rate.

- **F5-TTS**[10] [51] is a non-autoregressive TTS system based on Diffusion Transformer (DiT) [64] and flow matching [65]. It forgoes duration models and alignment by padding text to match speech length, using ConvNeXt V2 [66] to refine text features. An inference-time Sway Sampling strategy improves decoding efficiency without retraining. Trained on a 100K-hour multilingual dataset, F5-TTS supports zero-shot synthesis, expressive speech generation, speed control, and seamless code-switching.

---

[7]`https://github.com/HLTSingapore/Emotional-Speech-Data`

[8]`https://github.com/index-tts/index-tts`

[9]`https://github.com/SparkAudio/Spark-TTS`

[10]`https://github.com/SWivid/F5-TTS`

- **CosyVoice2** [37] is a language model-based TTS system designed for zero-shot control of both emotion and speaker identity. Further architectural and training details are provided in Appendix A.

All baseline systems share the same inference procedure: each sentence is divided into multiple word-level segments with specified emotional states and speaking rates. These segments are synthesized separately using emotion cloning combined with their respective speaking rate control strategies, and then concatenated to form the final speech.

## G.3 Evaluation Metrics

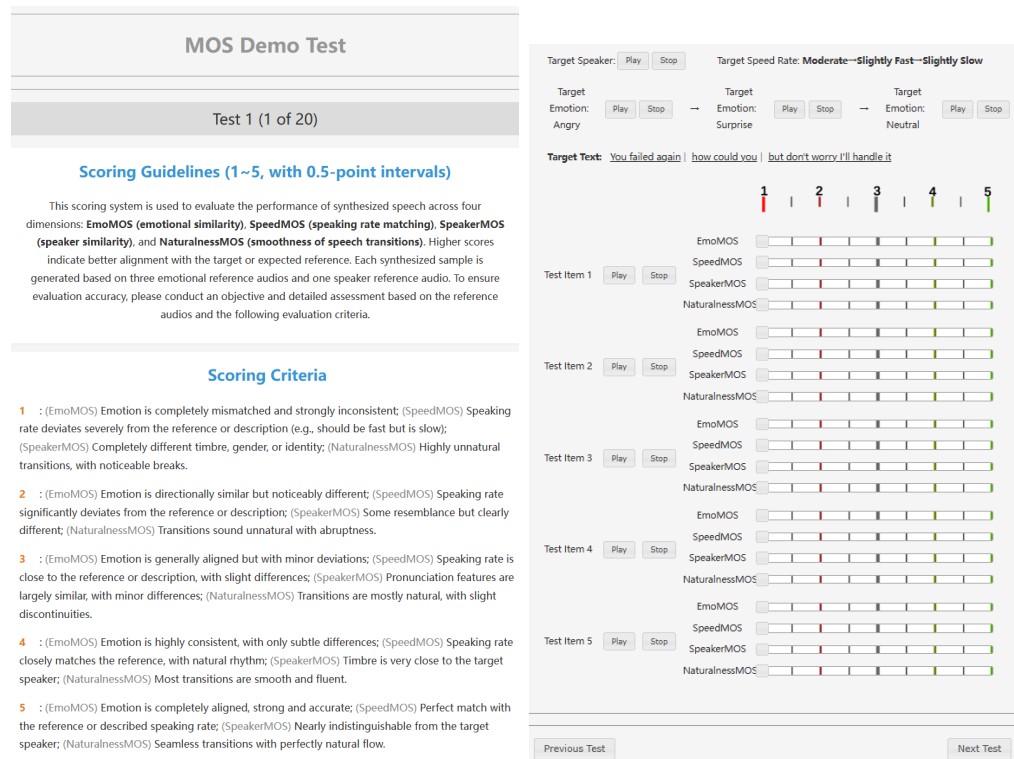

Figure 9: The MOS evaluation interface used for rating emotion consistency, speaking rate consistency, speaker similarity, and transition smoothness.

**Objective Metrics** The objective evaluation is conducted in two groups: **Group 1.** Given the generated speech, the target speaker's prompt, and the reference transcript, we compute three utterance-level metrics: character accuracy, speaker similarity, and DNSV (the variance of DNSMOS-PRO[11] [59] scores). Character accuracy is computed by comparing the output of an automatic speech recognition (ASR) model against the target transcript. Specifically, we use Paraformer[12] [54] to calculate character error rate (CER) for Chinese and Whisper Large V3[13] to compute word error rate (WER) for English. Speaker similarity is measured by extracting utterance-level embeddings from the generated speech and the target prompt using WavLM-Large[14] [55], followed by computing the cosine similarity between them. DNSV is used to assess transition smoothness. DNSMOS-PRO scores are calculated over the generated speech using a 2-second window and a 1-second stride. The

---

[11]https://github.com/fcumlin/DNSMOSPro
[12]https://github.com/modelscope/FunASR
[13]https://github.com/openai/whisper
[14]https://github.com/microsoft/UniSpeech/tree/main/downstreams/speaker_verification

variance of these scores is used to quantify transition smoothness, with higher variance indicating lower smoothness. Since the value of the variance is often relatively small, we multiply it by 100 for display purposes. **Group 2.** Based on the ASR transcription obtained in Group 1, we perform forced alignment to determine word-level timestamps. A string-matching strategy is then used to align each generated word-level segment with its corresponding emotional prompt, according to the original text-emotion-speed mapping. For each aligned pair, to evaluate expressive similarity, the emotional prompt is first adjusted to the target speaking rate using a phase vocoder algorithm[15] [67]. The generated segment is then compared to the rate-adjusted prompt using AutoPCP[16] [56] to compute prosodic similarity. Emotion embeddings are extracted using emotion2vec-large[17] [57] and a wav2vec-based model[18] [58], and cosine similarity is calculated to quantify emotion similarity.

**Subjective Evaluation**     We conduct Mean Opinion Score (MOS) evaluations from four perspectives: emotional consistency, speaking rate consistency, speaker similarity, and smoothness of emotional transitions. For each aspect, we provide participants with detailed evaluation criteria and report both the mean scores and 95% confidence intervals. A total of 15 graduate students with research backgrounds in speech emotion recognition or emotional speech synthesis participated in the evaluation. Prior to the test, all participants were provided with a detailed explanation of the interface and task. They were also informed that the data would be used for scientific research purposes. Each participant rated 20 sets of results (10 in Chinese and 10 in English) generated by five different systems. The complete evaluation took an average of approximately 49 minutes per participant. Scores were assigned on a 1 to 5 scale with 0.5-point intervals. The evaluation interface is shown in Figure 9.

# H    Out-of-Domain Evaluation

To evaluate the generalization ability of our approach under an out-of-domain dataset, we conduct word-level emotion and speaking rate control experiments on the CASIA dataset [68], as organized according to Appendix G. The CASIA corpus is a Mandarin emotional speech dataset recorded by four native speakers and covers six emotion categories: neutral, angry, fear, happy, sad, and surprise. Some of these emotions are not seen during training, which makes CASIA suitable for testing the cross-domain robustness of controllable speech synthesis. The results are shown in Table 6. Our method, WeSCon, demonstrates strong performance across nearly all evaluation metrics, achieving lower DNSV and higher speaker similarity (S-SIM), emotional similarity (Emo2vec), and arousal scores compared to other baselines. The overall results are consistent with those in Table 1, further confirming that our method generalizes well to unseen speakers and novel emotional patterns.

Compared to Table 1, the student model (WeSCon 2nd) shows slightly weaker performance than the teacher model on the out-of-domain test set, in some metrics. This degradation is primarily caused by the data filtering strategy adopted during self-training, which improves performance on in-domain speakers and emotions but may introduce subtle biases, resulting in mild overfitting. Nevertheless, such performance fluctuations are acceptable given that the second-stage model significantly simplifies the inference process.

Table 6: Objective evaluation results on the CASIA-based evaluation dataset for word-level emotion and speaking rate control.

| Method | WER/CER↓ | DNSV↓ | S-SIM↑ | AutoPCP↑ | Emotion↑ Emo2v. | Aro. |
|---|---|---|---|---|---|---|
| Index-TTS | **1.217** | 8.887 | 0.468 | 2.444 | 0.824 | 0.502 |
| F5-TTS | 1.374 | 8.940 | 0.462 | 2.539 | 0.845 | 0.526 |
| Spark-TTS | 1.299 | 8.720 | 0.439 | 2.496 | 0.841 | 0.523 |
| CosyVoice2 | 1.405 | 8.093 | 0.542 | 2.503 | 0.835 | 0.517 |
| WeSCon (1st) | 1.411 | 4.680 | 0.587 | **2.670** | **0.869** | 0.548 |
| WeSCon (2nd) | 1.478 | **4.641** | **0.590** | 2.624 | 0.867 | **0.552** |

[15]https://librosa.org/doc/latest/generated/librosa.effects.time_stretch.html#librosa-effects-time-stretch

[16]https://github.com/facebookresearch/stopes/blob/main/stopes/eval/auto_pcp

[17]https://github.com/ddlBoJack/emotion2vec

[18]https://github.com/audeering/w2v2-how-to

# I    Training Progress

We present the evolution of key validation metrics throughout the two-stage training process, as illustrated in Figure 10 and Figure 11. Figure 10 displays the frame-level accuracy of the aligner model in the first stage, covering both text token prediction and boundary detection. Figure 11 reports the accuracy of speech token prediction and the frame-level emotion prediction by the emotion aligner in the second stage. As shown, the aligner consistently achieves high frame-level accuracy in both stages. This is expected, as the target classes for both text and emotion are provided as input, and the aligner's primary objective is to learn accurate alignments, which is a relatively straightforward task given the model's underlying text-to-speech capabilities.

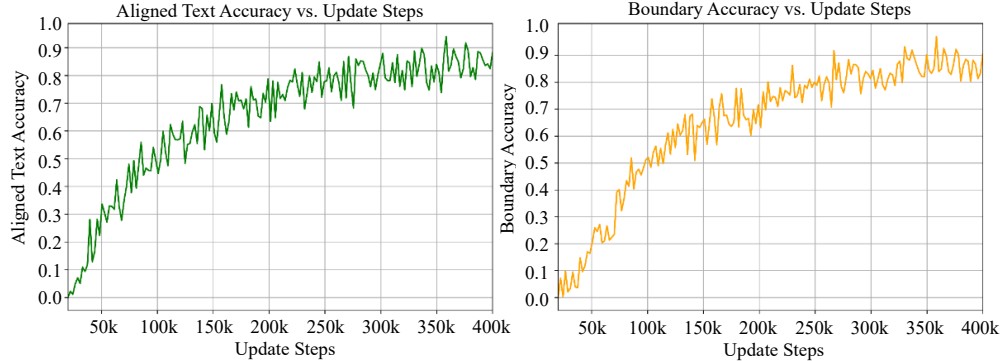

Figure 10: Validation accuracy of frame-level text token and boundary prediction by the aligner during the first stage training.

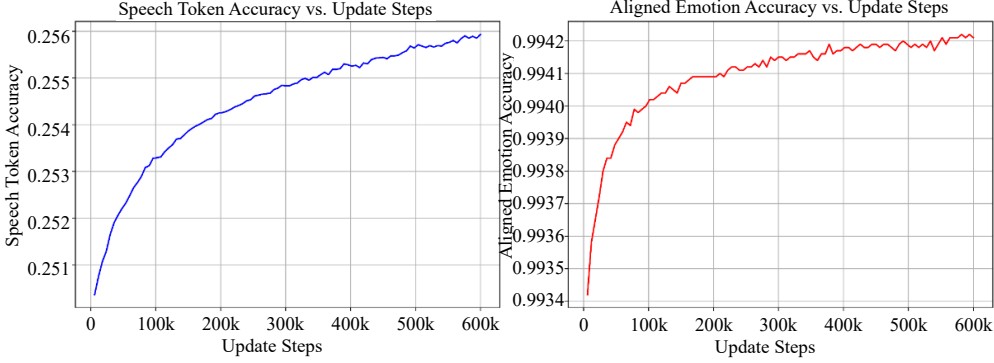

Figure 11: Validation accuracy of speech token prediction and aligner's frame-level emotion label prediction during the second stage self-training.

