# OpenReview forum: "Word-Level Emotional Expression Control in Zero-Shot Text-to-Speech Synthesis"
_NeurIPS.cc/2025/Conference — NeurIPS 2025 spotlight_

### Official Review · Reviewer_5szS · 2025-06-18

**Clarity:** 4
**Significance:** 4
**Originality:** 4
**Rating:** 5
**Confidence:** 5

**Summary:**

The paper introduces WeSCon (Word-level Emotion and Speed Control for TTS), a novel two-stage framework to achieve word-level emotion and speaking rate control using a pre-trained zero-shot TTS model (CosyVoice 2) without an explicitly annotated dataset, i.e, without fine-grained emotional information. In Stage one, they train a content aligner (supervised by the output of a forced aligner) and use a multi-inference method with different dynamic speed control mechanisms (interpolating prompts to slow or faster the speaking rate), and a Transition-smoothing strategy (tail-to-head linkage) to guide the model to do word-level expressive synthesis through multi-round inference. However, this method requires multiple inference steps which can be time consuming, thus, they use a teacher-student self training paradigm with Dynamic Emotional Attention Bias (DEAB) (essentially predicting weight for 7 different predefined attention masks during generation) and emotion prediction resulting in end-to-end word-level emotion and speed control for TTS with minimal to no performance drop.

**Questions:**

None

**Ethical Concerns:**

["NO or VERY MINOR ethics concerns only"]

**Final Justification:**

The authors addressed my questions and pointed out weaknesses.

**Limitations:**

Yes

**Quality:**

4

**Strengths And Weaknesses:**

### Strength
- The paper is well written, logically structured and most design choices are supplemented with ablations resulting in a very through experimentation.
- The paper requires a small utterance-level annotated dataset and can still achieve good quality word-level emotion and speed control, which is especially useful, as the authors mention that such datasets can be expensive to obtain or annotate.
- The model consistently outperforms strong baselines, and the effect of the dynamic speed control mechanism, transition smoothing, and DEAB is also objectively ablated, showing their benefits clearly.
- Again, for the experiments, a comprehensive and diverse set of objective metrics were used to evaluate all the models, providing robust evaluations.

### Weakness
* The subjective evaluations have overlapping confidence intervals, resulting in insignificant differences between comparing systems because of a small number of evaluators.
* Limited emotional granularity (mainly because of available datasets) as we can sample only from a discrete number of emotions and not walk in a continuous space.
* Training overhead, to train despite of use of a small dataset, one has to do multi-round inference, do force alignment, generate synthetic data (from an LLM), train CosyVoice 2, etc. However, the stage 2 training helps mitigate it.

---

> ### Author Rebuttal · Authors · 2025-07-30
>
> We are very encouraged by your recognition of our paper's clarity, technical design, and comprehensive experimentation. We are also grateful for your constructive observations, which help us further clarify our contributions and limitations. Below we respond to each of the concerns raised.
>
> - W1: On subjective evaluation confidence intervals.
>
> > A1: Thank you for the observation. While some subjective scores, such as EMOS, show overlapping confidence intervals, this is common in MOS evaluations with a limited number of raters. In our case, 15 professionally trained evaluators assessed 20 comparison groups, each consisting of 5 samples. This setup represents a typical configuration that balances statistical reliability and practical cost. Overlap in confidence intervals does not necessarily imply that the systems are indistinguishable, particularly when the mean scores consistently favor our model across all perceptual dimensions, including SPMOS, SMOS, NMOS, and EMOS.
> >
> > These trends are also strongly supported by objective results in Table 1, where our model outperforms all baselines across expressive metrics. The ablation studies further confirm that each component contributes to measurable gains in both control and expressiveness.
>
> - W2: On limited emotional granularity, as we can sample only from a discrete number of emotions and not walk in a continuous space.
>
> > A2: We appreciate the reviewer’s insightful comment. While our test set is constructed using discrete emotion labels, the model is not inherently restricted to fixed emotional categories. When the prompt speech contains natural emotional variation, including subtle or gradual shifts, the model is able to capture and reproduce these changes in a coherent and expressive manner.
> >
> > As shown on our demo page, the system effectively handles prompts that include emotionally diverse segments, whether from different speakers or from the same speaker expressing different emotional tones. The resulting speech demonstrates smooth transitions and appropriate emotional flow, suggesting that the model can respond flexibly to emotional variation in realistic scenarios.
> >
> > The use of discrete labels in our evaluation was mainly intended to support large-scale testing and controllable supervision, in light of the lack of datasets annotated with continuous or intra-sentence emotional changes. We agree that finer-grained or continuous emotional modeling is a valuable direction, and we believe our framework offers a strong foundation for future extensions in this area.
>
> - W3: On training overhead. However, the stage 2 training helps mitigate it.
>
> > A3: We appreciate the reviewer’s thoughtful observation. While our two-stage self-training framework introduces additional steps, including multi-round inference, forced alignment, and synthetic data generation, this design is primarily motivated by the lack of labeled emotion-transition data. The first stage plays a crucial role in constructing pseudo-supervised signals that activate word-level expressive control in a pretrained zero-shot TTS model. As the reviewer rightly noted, the second-stage student model successfully absorbs this supervision into an end-to-end architecture. This not only simplifies inference but also mitigates potential cascading errors from the teacher pipeline while maintaining controllability.
> >
> > In terms of training overhead, although the overall pipeline involves several components, each step is relatively efficient. For example, as shown in Appendix H, the content aligner can be trained quickly with a small amount of data, due to the strong text-speech alignment capabilities of pretrained TTS models. Additionally, the LLM-based script generation is not time-consuming, as the total number of emotion scripts required remains manageable even when covering thousands of hours of speech. Finally, the inference process of the teacher model can be accelerated through parallelization. With eight A100 GPUs and vLLM support, the system can synthesize approximately 40 hours of supervision data per hour, enabling efficient data generation at scale.

---

> > ### Comment · Reviewer_5szS · 2025-08-05
> > **Comments have been addressed**
> >
> > Thank you for your further clarification. My comments have been addressed.

---

> > > ### Author Response · Authors · 2025-08-07
> > > **Appreciation for the 5szS's Feedback.**
> > >
> > > Thank you for your response and for acknowledging that our clarifications addressed your comments. We truly appreciate your recognition of our work.

---

### Official Review · Reviewer_zpVF · 2025-06-30

**Clarity:** 2
**Significance:** 3
**Originality:** 3
**Rating:** 4
**Confidence:** 4

**Summary:**

This paper proposes the WeSCon framework, which achieves end-to-end word-level emotional expression control for the first time without requiring emotion-annotated data, through a self-training framework and a dynamic emotional attention bias mechanism. The method generates mixed-emotion training data (combining speech segments with different emotional prompts) via a multi-round inference mechanism and designs a dynamic emotional attention bias module to enable pretrained TTS models to support fine-grained word-level emotion and speech rate joint control. Experiments demonstrate that WeSCon achieves state-of-the-art (SOTA) performance in word-level emotional expression under limited data while preserving the zero-shot generation capability of the pretrained model, offering a novel approach for emotional speech synthesis in low-resource scenarios.

**Questions:**

see Weakness

**Ethical Concerns:**

["NO or VERY MINOR ethics concerns only"]

**Final Justification:**

After reading other reviewers' comments and your rebuttal, I have decided to improve my rating to 4, a positive score. I agree that the emotional interpretation does not always follow directly from the literal wording. I think the proposed method is a promising way to synthesize speech with given dynamic emotional expressions.

**Limitations:**

see Weakness

**Quality:**

3

**Strengths And Weaknesses:**

Strengths:

1. Emotional speech synthesis is a critical direction in the field of text-to-speech (TTS), and data scarcity remains a key challenge. This work addresses emotion control without relying on emotion-annotated data, demonstrating notable application value and research significance.

2. The multi-round inference mechanism generates mixed-emotion training data by combining speech with different emotional prompts, eliminating the need for large-scale emotion-annotated data and improving data utilization efficiency. The dynamic emotional attention bias mechanism enables pretrained TTS models to perceive emotion boundaries and speech rate variations, supporting joint control of word-level emotion and speech rate during inference, thereby enhancing expressive capability.

3. The experiments verify the effectiveness of the proposed method in word-level emotional expression control. Comparative results with baseline models demonstrate its performance advantages. Ablation studies further validate the contributions of individual components, reflecting the rationality of the design.

Weaknesses:

1. While high-expressive emotional speech synthesis is an important research direction, the baseline selection in the experiments exhibits bias. Compared models (e.g., CosyVoice, Spark-TTS, F5-TTS) are not specifically designed for emotional synthesis. The sentence-level concatenation strategy may unfairly advantage the proposed method in comparisons, as baseline models generate utterances holistically without boundary artifacts. This asymmetry could affect performance evaluation.

2. The current implementation adjusts speech rate merely by upsampling/downsampling speech tokens without explicitly linking it to emotional expression features (e.g., stress, rhythm, prosodic boundaries). This decoupled design may result in acoustically similar attributes for the same emotion at different speech rates, contradicting linguistic principles (where emotions should exhibit distinct prosodic patterns at varying rates). Further analysis of the emotion-speech rate interaction or supplementary experiments is recommended.

3. The reliance on GPT-4o for generating text and emotion labels introduces definitional ambiguity. The "reasonable emotion labels" lack clear criteria. If "reasonable" refers to statistical consistency with common scenarios, labels may overfit textual semantics (as modern LLMs can infer basic emotion shifts from context).

4. In my opinion, the application of this proposed method is limited. The acoustic features of the generated speech should come from the semantic understanding of the given text rather than a cherry-picked, complicated prompt waveform. For the scenario where the given emotion prompt conflicts with the semantic information of the given text, this paper did not provide an in-depth analysis. To me, this paper is fancy but lacks practical applicability.

---

> ### Author Rebuttal · Authors · 2025-07-30
>
> We sincerely thank the reviewer for the detailed and thoughtful feedback. We appreciate your recognition of the novelty, technical design, and experimental rigor of our work, as well as the practical significance of word-level emotional expression control under low-resource conditions. Below we respond point-by-point to the concerns raised.
>
> - W1： On the fairness of baseline comparisons.
>
> > A1: We believe our comparisons are fair and representative.While CosyVoice2, Spark-TTS, IndexTTS, and F5-TTS are primarily designed for general-purpose zero-shot TTS, they are among the state-of-the-art open-source zero-shot TTS models, and all offer strong emotion cloning capabilities. In contrast, open-source models specifically designed for emotional synthesis (e.g., EmotiVoice) do not support speaker cloning, making direct comparisons under a unified evaluation protocol infeasible.
> >
> > Furthermore, as shown in Table 3, our focus is to preserve the strong zero-shot generalization of pretrained TTS while enabling fine-grained expressive controlling. While we acknowledge that differences in model capabilities may introduce some asymmetry, we have followed a consistent evaluation protocol and believe the selected baselines are the most relevant and competitive open-source alternatives for assessing fine-grained emotional expression.
>
> - W2: On the interaction between speaking rate and emotion.
>
> > A2: This is an excellent point. We agree that speech rate and emotion are closely interrelated, and emotional states often exhibit distinct prosodic patterns depending on speaking tempo. We would like to clarify that our method does not assume independent control of speech rate and emotional expression. In fact, our findings support the hypothesis that the two are interdependent. Since both speaking rate and emotional features are provided through the same prompt speech, adjusting the rate through upsampling or downsampling inevitably affects emotional expression as well.
> >
> > As shown in Table 4 (row 3), removing speech rate control leads to noticeable drops in both emotional expression similarity and naturalness (During evaluation, we adjust the target speech to match the target speaking rate before computing emotion similarity, see Appendix F.3, line 769).
> >
> > To further investigate the effect of speaking rate on perceived emotional expression, we conducted an additional experiment (will be added in the appendix). We randomly selected 100 emotional utterances from the test set as prompts and systematically varied the resampling ratio from 0.5 to 2.0 in steps of 0.25, using the same target text (this range was chosen to ensure performance stability; see Appendix B). For each result, we computed the emotion similarity between the generated speech and both the original and rate-adjusted (re-rated) target using Emotion2Vec Score (Emition similarity):
> >
> > | Resampling Ratio | Emotion2Vec Score ↑ | Emotion2Vec Score (re-rated) ↑ |
> > | --- | --- | --- |
> > | 0.5 (Downsampled to half, speed up) | 0.57 | 0.86 |
> > | 0.75 | 0.85 | 0.88 |
> > | 1.0 | 0.90 | 0.90 |
> > | 1.25 | 0.83 | 0.89 |
> > | 1.5 | 0.75 | 0.90 |
> > | 1.75 | 0.68 | 0.91 |
> > | 2.0 (Interpolated to twice, speed down) | 0.51 | 0.87 |
> >
> > The results show that emotional expression similarity drops significantly if the reference is not adjusted for speaking rate, while similarity remains stable when compared against rate-matched references. This confirms that speech rate provides important prosodic cues for emotional expression, consistent with our findings in Section 4.2.2.
>
> - W3: On concerns about GPT-4o-generated text and emotion labels.
>
> > A: We appreciate the reviewer’s insightful question. In our paper, *“reasonable emotion labels”* refer to emotional changes that fit the scenario, context, and character relationships defined in the prompt. Rather than assigning labels manually, we let GPT-4o generate emotional expressions based on structured input such as background, context, and role pairs.
> >
> > To reduce semantic overfitting, we independently prepared over 2000 options for each of backgrounds, contexts, and roles, enabling up to 8 billion combinations through free permutation. This diversity helps guide GPT-4o to produce contextually grounded emotion shifts.
> >
> > We acknowledge that the diversity of generated data is still limited by the LLM’s capacity and the size of our emotion prompt dataset, as discussed in Figure 5. Fully addressing this would require richer emotional corpora, which is beyond the scope of this work. In addition, since emotional planning in our method is determined by the sequence of emotion labels, but the detail expressiveness is primarily driven by the emotional speech prompts, our framework does not heavily rely on GPT-4o as a source of emotional intent. For instance, such intent can also be derived from narrative texts such as novels or scripts.
> >
> > Our goal is to show that, even without emotion-transition data, a pretrained zero-shot TTS model can learn fine-grained expressive control through self-training. We believe that a detailed discussion of GPT-4o’s performance is beyond the scope of our paper.
>
> - W4: On the applicability and practicality of the proposed method and the possible mismatch between prompt emotion and text semantics.
>
> > A: We appreciate the reviewer’s concern regarding the practicality of our method and the possible mismatch between prompt emotion and text semantics.
> >
> > As the reviewer pointed out, emotional expression should reflect the semantic meaning of the text. We fully agree with this view. Our method assumes that the emotional intent has already been determined based on the semantic content, whether through human interpretation or external models (GPT-4o). The speech prompt provides expressive cues such as prosody, rhythm, and speaking rate. These cues help to deliver the intended emotion more effectively and work together with the semantics rather than replacing it.
> >
> > Regarding the possible mismatch between prompt speech emotion and text meaning, we believe such cases often reflect the gap between literal text and the speaker’s intended meaning in context. Emotional interpretation does not always follow directly from the literal wording. For example, the sentence "What a beautiful day" may be used sincerely or sarcastically depending on the situation. Similarly, "I'm fine" can express calmness, annoyance, or resignation. In these cases, a prompt that seems emotionally different from the surface text can still match the speaker's true intent. Therefore, a surface-level mismatch does not necessarily mean a conflict with semantics.
> >
> > How the emotional intent is obtained, whether through semantic analysis, narrative planning, or other upstream processes, is not the focus of our work. Our goal is to generate word-level expressive speech once the emotion and text are given. Our experiments show that this framework produces natural and coherent speech across a range of emotional contexts. This supports its practical value in real-world applications such as audiobook synthesis, dialogue systems, and storytelling.

---

> > ### Comment · Reviewer_zpVF · 2025-08-04
> >
> > Thanks for your responses. After reading other reviewers' comments and your rebuttal, I have decided to improve my rating to 4, a positive score. I agree that the emotional interpretation does not always follow directly from the literal wording.  I think the proposed method is a promising way to synthesize speech with given dynamic emotional expressions.

---

> > > ### Author Response · Authors · 2025-08-07
> > > **Appreciation for the zpVF's Feedback.**
> > >
> > > We sincerely appreciate your thoughtful response and your decision to raise the rating of our submission. We are especially grateful for your recognition of our work. Thank you again for your time and consideration.

---

### Official Review · Reviewer_LGsJ · 2025-07-03

**Clarity:** 4
**Significance:** 4
**Originality:** 4
**Rating:** 6
**Confidence:** 4

**Summary:**

This paper proposed a word-level emotion control-based zero-shot TTS. Instead of using utterance level emotion prompt to derive the emotional speech, this paper proposed a multi-stage inference method that the word-level emotion embeddings and speaking rate is automatically learnt which were used to help to predict the proper expressive speech. Furthermore, to achieve a end-to-end system, this paper adopted the self-training scheme to use the multi-stage system as a teacher to guide the end-to-end model training. Experiment and results show the proposed method is very promissing in terms of both objective and subjective evalution metrics.

**Questions:**

Can the emotion transfer across language? For example, if you use English speech as prompt and generate Chinees speeh, can the proposed method still generate controlled expressive speech?

**Ethical Concerns:**

["NO or VERY MINOR ethics concerns only"]

**Final Justification:**

This paper proposed an innovative word-level emotion control-based zero-shot TTS rather than a conventional utterance-based method. The method is sounding and the results are promissing. In the rebutal stage, the authors clarfied two questions I rasied about generated speech speed and cross lingual emotion transfer, which are reasonable and clear. Based on these, I will keep my original score which is 6-Strong Accept.

**Limitations:**

yes

**Quality:**

4

**Strengths And Weaknesses:**

Strengths
1, This paper is addrssing a challenging problem of zero-shot expressive text to speech syntheis with an innovative method, which is word-level emotion control. This method has advantage over the utterance-based prompt method as it can have a more precise emotion control, particular for generating speech with changing tone in a single sentence.
2, The paper is very well-written with a clear clarity.
3, The techniques are sounding.
4, The experiments are comprehensive and the results are very promissing compared to the SOTA methods in the expressive TTS field. The sound quality and naturalness of the samples in the demo page is very impressive.

Weakness
One of the small issue is that for the speech rate control, the paper proposed to do either downsampling or nearest-neigbor interpolation. However, the paper did not mention any method to predict the speech length. There is a concern that if you downsample or interpolate the speech samples with an arbitary factor, it could generate unnatural speech.

---

> ### Author Rebuttal · Authors · 2025-07-30
>
> We sincerely thank you for your positive and encouraging feedback. We are glad that you found our method for word-level emotion control to be innovative and effective, especially in addressing the challenge of generating expressive speech with intra-sentence variation. We also appreciate your recognition of the writing quality, technical soundness, and the strength of our experimental results and audio demos. Below we respond to the specific concern and question raised:
>
> - W1: The paper proposes using downsampling or interpolation to adjust speech rate, but does not describe how target durations are determined. Arbitrary rescaling may result in unnatural speech.
>
> > A1: Thank you for highlighting this important detail. In our system, the speaking rate is not arbitrarily defined but computed through a structured process informed by both LLM outputs and statistics of the non-emotion-transition training dataset:
> >
> > 1. During supervision generation of teacher model, GPT-4o specifies the desired speaking rate for each word (ranging from 0.5 to 2.0, where 1.0 denotes normal speed), as part of the scripted prompt generation (see Appendix C).
> >
> > 2. For each emotion category, we calculate the average phoneme duration (i.e., average duration per phoneme) from non-emotion-transition training data.
> >
> > 3. The target phoneme duration is obtained by multiplying the LLM-specified rate with the emotion-specific average duration.
> >
> > 4. We then compare this target to the phoneme duration of the prompt speech, and apply either linear interpolation or downsampling accordingly.
> >
> >
> > This approach ensures that duration modification is linguistically informed, emotion-sensitive, and statistically regularized. In experiments, we observed that this method preserves the naturalness and rhythm of generated speech without introducing artifacts. We will include this implementation detail in the appendix of the final version.
>
> - Q1: Can the emotion transfer across language? For example, if you use English speech as prompt and generate Chinese speeh, can the proposed method still generate controlled expressive speech?
>
> > QA1: Our backbone model, CosyVoice2, supports cross-lingual zero-shot TTS. In our preliminary experiments, the first-stage WeSCon model successfully handles cross-lingual intra-sentence emotional variation. The second-stage model can also generate expressive speech in cross-lingual settings, but exhibits noticeable accent-related distortions. This is mainly due to the fact that the student model in Stage 2 was not explicitly trained with cross-lingual supervision or data organization.
> >
> > Theoretically, the student model can achieve comparable cross-lingual performance by incorporating structured multilingual self-training data. However, this extension falls beyond the scope of the current paper. We consider this a promising direction and plan to explore it in future work.

---

> > ### Comment · Reviewer_LGsJ · 2025-08-06
> >
> > Thanks for the detailed explanation, which sounds clear to me now.

---

> > > ### Author Response · Authors · 2025-08-07
> > > **Appreciation for the LGsJ's Feedback.**
> > >
> > > We sincerely appreciate your kind response and your recognition of our work. Thank you very much for your time and support.

---

### Official Review · Reviewer_uPUi · 2025-07-05

**Clarity:** 3
**Significance:** 3
**Originality:** 3
**Rating:** 4
**Confidence:** 3

**Summary:**

The paper introduces WeSCon, a novel two-stage self-training framework for enabling word-level control over both emotion and speaking rate in a zero-shot text-to-speech (TTS) model. The primary challenge addressed is the scarcity of datasets with fine-grained, intra-sentence emotional annotations. Experimental results, on both English and Chinese, demonstrate that WeSCon achieves state-of-the-art performance in word-level expressive control. It significantly improves transition smoothness and emotional accuracy compared to strong baselines, all while preserving the original model's zero-shot speaker cloning capabilities.

**Questions:**

The paper's core design choice—a teacher-student framework—is justified by the need for a more efficient inference process. This claim is currently unsubstantiated. Can you provide a quantitative comparison of inference speed (e.g., Real-Time Factor) for: (a) the teacher model, (b) the final student model, and (c) the baseline models. Without this data, the central motivation for the entire second stage of the work is unconvincing.

**Ethical Concerns:**

["NO or VERY MINOR ethics concerns only"]

**Final Justification:**

Thanks for reviewers' rebuttal, it solve my concerns, I have raised my point to 4.

**Quality:**

3

**Strengths And Weaknesses:**

Strengths
- The paper addresses an important challenge in TTS. Achieving fine-grained, word-level control over expressive attributes is a logical and necessary next step for creating more natural-sounding synthetic speech.
- The paper is generally well-written and structured. The figures and descriptions of the architecture provide a clear high-level understanding of the proposed pipeline.

Weaknesses
- The entire self-training process is critically dependent on texts generated by a proprietary model GPT-4o. In other words, the system can only learn emotional transitions that the LLM can imagine and articulate, potentially limiting its diversity and introducing stylistic artifacts.
- The title claims "Word-Level Emotional Expression Control," which suggests a high degree of flexibility. However, the control is limited to a small, predefined set of discrete emotions, and the transitions are pre-scripted by LLMs. The system does not model the evolution of emotion (e.g., a gradual shift from surprise to anger) but rather concatenates discrete emotional states. This is a significant limitation on the naturalness of the expression and may not generalize to more complex, real-world emotional dynamics.
- A primary motivation for the complex two-stage design is to overcome the inefficiency of the teacher's multi-round inference. However, the paper does not show any quantitative analysis of inference speed (e.g., Real-Time Factor). Without this result, it is impossible to judge whether the student model provides a meaningful practical benefit. The entire premise of the second stage rests on an unproven assumption of efficiency gain.

---

> ### Author Rebuttal · Authors · 2025-07-30
>
> We thank you for your thoughtful feedback and constructive criticisms. We appreciate that you acknowledged the significance of fine-grained expressive control in TTS, and the clarity of our proposed pipeline. Below, we respond to the main concerns raised.
>
> - W1: Reliance on GPT-4o-generated text may limit the diversity and naturalness of emotional
>   transitions.
>
> > A1: We agree that relying on GPT-4o-generated text introduces limitations in the diversity and naturalness of emotional transitions. This concern is valid and has been explicitly acknowledged in the Limitations section of our paper.
> >
> > Our goal is not to evaluate the LLM itself, but to show that a TTS model can achieve word-level expressive control using only non-switching emotion data. In our framework, GPT-4o serves solely as a tool to create pseudo-supervision for transitions absent in the original dataset.
> >
> > To improve the diversity of generated expressions, we independently constructed over 2000 options for each of dialogue settings, contextual situations, and character relationships, allowing for up to 8 billion unique combinations. GPT-4o was prompted with carefully crafted instructions to ensure appropriate emotional flow. For example, the first case on our demo page involves a formal discussion between a leader and an employee about project progress.
> >
> > While GPT-4o may not fully capture the richness of real-world transitions, and this may partly explain our method's performance bottleneck (see Figure 5 in our paper). Our results show that the model still learns to generate smooth and perceptually natural emotional shifts. This demonstrates the practical effectiveness of our approach despite GPT-4o's limitations. In addition, since emotional planning in our method is determined by the sequence of emotion labels, but the detail expressiveness is primarily driven by the emotional speech prompts, our framework does not heavily rely on GPT-4o as a source of emotional intent. For instance, such intent can also be derived from narrative texts such as novels or scripts.
>
> - W2: Discrete emotion concatenation limits the naturalness and flexibility of expression.
>
> > A2: We appreciate the reviewer’s concern and would like to clarify that our method does not inherently suffer from the issue described. When the speech prompt contains different emotions, the model is able to handle such emotional switches and generate coherent and natural speech. Our framework does not rely on artificial concatenation and supports expressive control across distinct emotional states.
> >
> > As demonstrated in our demo page, the model supports speech prompts composed of sentences from different speakers as well as from the same speaker with different emotional tones. This shows that our system can successfully generate speech based on prompts containing emotional changes, even across sentences with varying styles or intensities. Therefore, when the prompt comes from a single utterance that includes an emotional switch, the model is equally capable of producing natural and consistent results.
> >
> > In our experiments, we constructed prompts using segments with discrete emotions due to the lack of datasets annotated with intra-sentence emotional switches. This allowed us to conduct evaluations at scale while simulating diverse emotional scenarios. Despite this limitation, the model still achieves high naturalness and fluency, as supported by both subjective and objective results in Table 1 and Table 2 (Both objective and subjective metrics achieve state-of-the-art performance).
>
> - W3&Q1: The paper lacks quantitative validation of the second-stage model’s efficiency on inference speed.
>
> > A3: The second-stage model is not designed to be faster than the first-stage pipeline. Instead, it brings several practical benefits, including simplified end-to-end inference, reduced error propagation from the multi-stage process, and improved transition consistency by leveraging full contextual emotional prompt, as shown in our experimental results.
> >
> > To address the reviewer’s concern, we conducted a quantitative comparison of inference speed under consistent conditions. All models were evaluated using the same test set (as in Table 1), on a single NVIDIA 2080Ti GPU. Results are shown below:
> >
> > **RTF** (Real-Time Factor): The ratio of total inference time (including prompt processing and target speech generation) to the duration of the generated speech. This metric reflects the real-time performance of the system when handling word-level expressive-controllable speech synthesis.
> >
> > **Token/s**: The number of tokens (including both the prompt and generated tokens) processed per second during inference. This reflects the architectural complexity.
> >
> > | Model | RTF | Token/s |
> > | --- | --- | --- |
> > | CosyVoice2 | 0.70 | 114.97 |
> > | WeSCon (1st) | 0.53 | 114.13 |
> > | WeSCon (2nd) | 0.68 | 109.38 |
> >
> > The first-stage model achieves a lower RTF than CosyVoice2, primarily due to fewer Flow Matching operations when handling multiple emotional transitions. The second-stage model introduces lightweight dynamic attention, which results in only a slight decrease in token/s, indicating that the overall architectural complexity remains similar. However, because the model attends to the all emotional prompts, the input sequence becomes longer during inference, leading to a more noticeable increase in RTF. Despite this, the inference speed remains comparable to CosyVoice2 while providing significantly better expressive smoothness and naturalness, as shown in Table 1 and Table 4. These results demonstrate the practical value of our self-training framework, even though inference efficiency is not its primary objective.

---

> ### Author Response · Authors · 2025-08-07
> **Kind Reminder: Clarification Before August 8 Rebuttal Deadline**
>
> Dear Reviewer uPUi,
>
> We hope this message finds you well. As the rebuttal period concludes on August 8, we would greatly appreciate it if you could let us know whether you have any further comments or concerns regarding our submission. Your feedback would be invaluable in helping us address any remaining issues.
>
> Thank you very much for your time and consideration.
>
> Best regards,
>
> The Authors

---

### Decision · Program_Chairs · 2025-09-17

**Decision:**

Accept (spotlight)

**Comment:**

After rebuttal, all reviewers recommend acceptance based on motivation, novelty and quality of results.